# The growth factor *BMP11* is required for the development and evolution of a male exaggerated weapon and its associated fighting behavior in a water strider

**William Toubiana**¤\*, **David Armisén**©, **Séverine Viala**, **Amélie Decaras**,
**Abderrahman Khila**© \*

Institut de Génomique Fonctionnelle de Lyon, Université de Lyon, Université Claude Bernard Lyon 1, Centre National de la Recherche Scientifique Unité Mixte de Recherche 5242, Ecole Normale Supérieure de Lyon, Lyon, France

¤ Current address: Department of Ecology and Evolution, University of Lausanne, Lausanne, Switzerland
\* toubianawilliam@gmail.com (WT); abderrahman.khila@ens-lyon.fr (AK)

**Data Availability Statement:** The raw sequencing data can be found in the BioProject ID: PRJNA610161 Scripts related to the analyses can

## Abstract

Exaggerated sexually selected traits, often carried by males, are characterized by the evolution of hyperallometry, resulting in their disproportionate growth relative to the rest of the body among individuals of the same population. While the evolution of allometry has attracted much attention for centuries, our understanding of the developmental genetic mechanisms underlying its emergence remains fragmented. Here we conduct comparative transcriptomics of the legs followed by an RNA interference (RNAi) screen to identify genes that play a role in the hyperallometric growth of the third legs in the males of the water strider *Microvelia longipes*. We demonstrate that a broadly expressed growth factor, Bone Morphogenetic Protein 11 (BMP11, also known as Growth Differentiation Factor 11), regulates leg allometries through increasing the allometric slope and mean body size in males. In contrast, *BMP11* RNAi reduced mean body size but did not affect slope either in the females of *M. longipes* or in the males and females of other closely related *Microvelia* species. Furthermore, our data show that a tissue-specific factor, Ultrabithorax (Ubx), increases intercept without affecting mean body size. This indicates a genetic correlation between mean body size and variation in allometric slope, but not intercept. Strikingly, males treated with *BMP11* RNAi exhibited a severe reduction in fighting frequency compared to both controls and *Ubx* RNAi-treated males. Therefore, male body size, the exaggerated weapon, and the intense fighting behavior associated with it are genetically correlated in *M. longipes*. Our results support a possible role of pleiotropy in the evolution of allometric slope.

## Introduction

Extravagant ornaments and weapons, often carried by males in a wide range of lineages, represent some of the most striking outcomes of directional sexual selection driven by female choice or male competition [1–4]. Exaggerated sexually selected traits are often characterized by their

be found in the Dryad link: https://datadryad.org/stash/share/2Q9tb7wD4R_iWkDSRAiOY2NsFuXCZE99o60C6Oeydk4.

**Funding:** This work was supported by an ERC Co-G WaterWalking #616346 and Labex CEBA to AK, and a BMIC Lyon PhD fellowship to WT. The funders had no role in study design, data collection and analysis, decision to publish, or preparation of the manuscript.

**Competing interests:** The authors have declared that no competing interests exist.

**Abbreviations:** BMP11, Bone Morphogenetic Protein 11; cDNA, complementary DNA; dsRNA, double-stranded RNA; GDF11, Growth Differentiation Factor 11; RNAi, RNA interference; Ubx, Ultrabithorax.

extreme growth and hypervariability among individuals of the same population [5,6]. These features derive from changes in scaling relationships, especially the elevation of the allometric coefficient (or allometric slope), which results in certain structures growing disproportionately larger relative to the rest of the body among individuals of the same population [5–8]. Variation in the size of morphological characters (Y) is often correlated with variation in body size (X) through a scaling relationship that follows a power law distribution such as $Y = aX_b$ [9,10]. When log transformed, this allometric equation becomes linear with *log(a)* representing the intercept, the relative size of trait Y, and *b* the slope, representing the proportional growth of trait Y relative to trait X. When b = 1, the 2 traits grow proportionally to one another, and when b is different from 1, there is a disproportionate growth between the 2 traits (hypoallometry with b < 1 or hyperallometry with b > 1). Despite the longstanding interest in the study of allometry, the developmental genetic mechanisms underlying covariation in growth patterns between traits and body among individuals of the same population are unclear [6,11–14].

Allometric slope is known to be relatively stable during evolution, and several studies suggest that this stasis could be the consequence of genetic correlation with other traits [9–13,15]. In the context of trait exaggeration (e.g., hyperallometry), Fisher predicted a genetic correlation between male exaggerated ornaments and female preference for the exaggeration—a process known as runaway selection [1]. Other studies, for example, in the ruffs or horned beetles, also suggest a genetic link between the elaboration of male exaggerated traits and other secondary sexual traits such as male body size and reproductive behavior [16–18]. However, empirical tests of these predictions remain difficult to achieve. Determining the genes that functionally influence slope and testing their correlation with other traits would therefore greatly advance our understanding of the evolution of scaling relationships [6,11–14].

We address these questions in the water strider *Microvelia longipes*, where some males exhibit extremely long third legs due to a hyperallometric relationship with body size [19]. Among the over 200 known species in this genus, *M. longipes* is the only species found where males simultaneously exhibit large body size and high variation in both body and leg length [19,20]. Males of other species that exhibit either large body, such as *Microvelia* sp. (see below), or high variation in body size, such as *Microvelia pulchella*, a sister species to *M. longipes*, do not display exaggerated leg length, suggesting that the evolution of hyperallometry may be linked to the evolution of increased mean and variation in body size [19]. In addition, males of *M. longipes* use the exaggerated legs to fight and dominate egg-laying sites where they intercept and mate with gravid females [19]. Compared to *M. pulchella*, *M. longipes* males fight over an order of magnitude more frequently, indicating that the evolution of the exaggerated weapon is also associated with the increase in competition intensity between males [19]. Here we use comparative transcriptomics combined with RNA interference (RNAi) as a tool to study gene function during development and determine the genes regulating the scaling relationship between the exaggerated legs and the body in *M. longipes* and 2 additional species. We found that the developmental gene *Bone Morphogenetic Protein 11* (*BMP11*), also known as *Growth Differentiation Factor 11 (GDF11)*, evolved a multifaceted role modulating the hyperallometry of the exaggerated legs, increased body size, and fighting intensity in males, thus revealing a genetic correlation between these traits in the male of *M. longipes*.

## Results

### Growth and gene expression differences in the legs during *M. longipes* nymphal development

In *M. longipes*, third leg length is variable among males and this variability is higher in males compared to females (Fig 1A and 1C) [19]. A developmental growth curve (Fig 1B; S1 Fig)

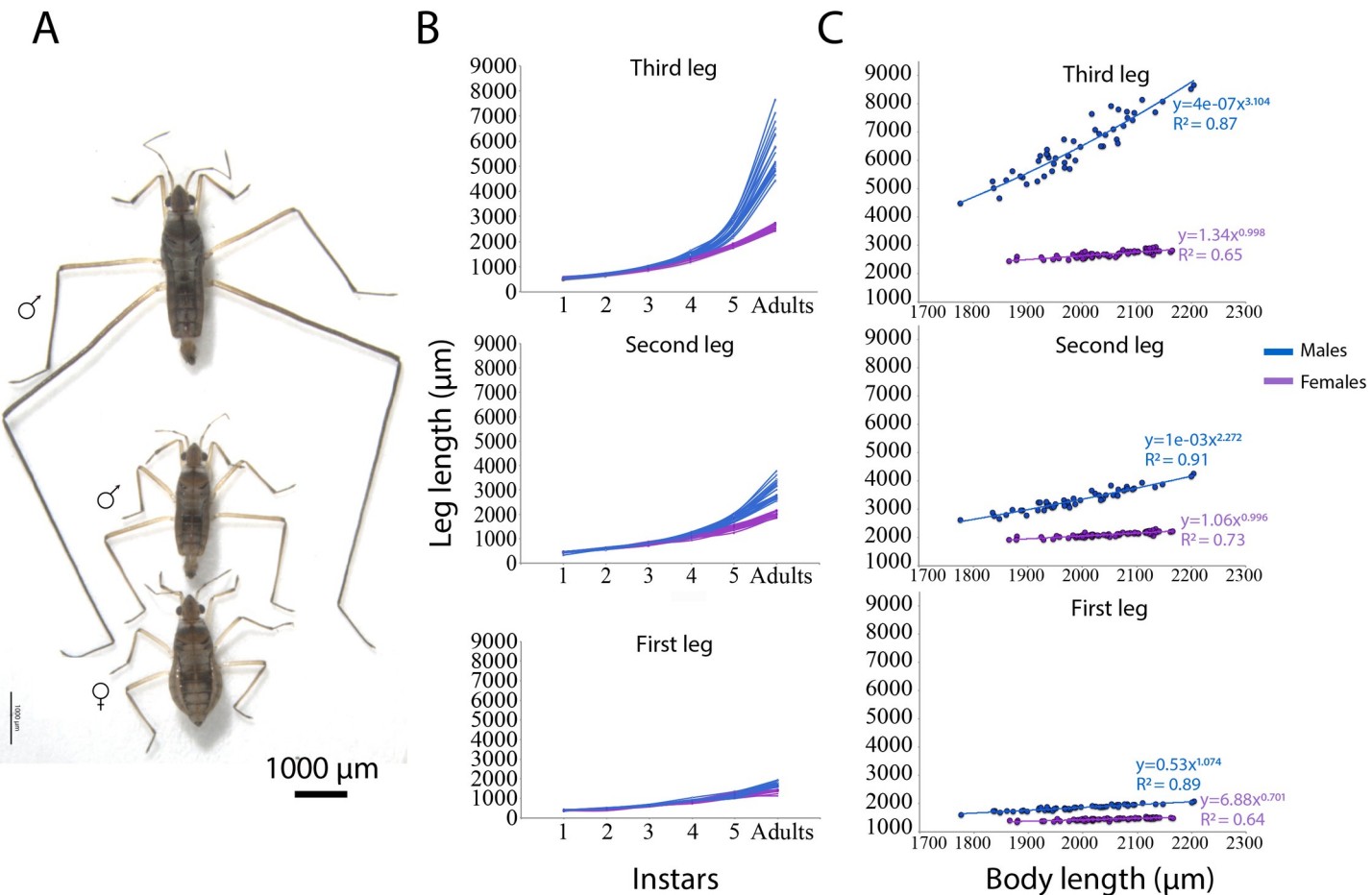

**Fig 1. Growth dynamics of male exaggerated third legs in *M. longipes*. (A)** Adult large and small males and a female showing final growth phenotypes. **(B)** Leg length variation during postembryonic development (nymphal development) in males and females. Each line represents leg growth dynamics of a single individual during nymphal development until adulthood. **(C)** Static allometries between body length and the 3 pairs of legs in adult males and females. Power law regressions were fitted to the raw data to represent the static allometric equation $y = bx^a$. Differences in static allometry parameters were tested on log-transformed data in [19]. The data underlying this figure may be found in S1 and S2 Tables.

revealed that leg length is similar between individuals at the first to third instars but starts to diverge between the sexes and also among males at the fourth instar (Fig 1B). The length of the third legs continues to increase between fifth instar and adult males in a higher rate when compared to females' legs or to the 2 other male legs during this last developmental stage (Fig 1B; S1 Table). These differences in leg length between individuals are independent of any variation in developmental time from first instar to adult (ANCOVA with body length as covariate, F-value = 0.156, *p*-value = 0.69; S2 Fig). Therefore, the growth burst and its variation at the end of postembryonic development in male exaggerated legs accounts for both the sexual dimorphism and the hypervariability among males (Fig 1A–1C). Ontogenetic allometry (i.e., scaling relationship between developmental stages of an individual) also confirmed the exaggerated growth rate of male third legs during development (S2 Table). The first and second legs in males and females also grew faster at the final instars, although their growth rate was not as extreme as the third legs (Fig 1B). Based on these observations, we hypothesize that the striking variation in leg growth dynamics within and between the sexes is attributable to variation in gene expression at the end of nymphal development.

Comparative transcriptomics of the legs at the fifth nymphal instar, the developmental stage where we detected a burst of growth, first revealed a global difference in gene expression profiles between the 3 pairs of legs in both sexes at this developmental window (S3A Fig). When comparing the first versus third legs, we found that the third legs diverged more from the first legs in males than they did in females, both in terms of number of differentially expressed genes and in the degree of differential expression (S3 and S4 Figs). Overall, these data indicate that leg exaggeration in *M. longipes* males is associated with a higher number and higher degree of differential expression of leg-biased genes in males compared to females. We used these datasets to select and test the role of candidate genes in leg exaggeration.

## Key developmental genes control the scaling relationship of male exaggerated legs

Extravagant signals and weapons, such as *M. longipes* third legs, are predicted to occur only in good condition individuals because of the high fitness cost they impose [4,21,22]. Proposed mechanisms for the development of exaggerated phenotypes include increased sensitivity or local production of growth factors by the exaggerated organ [4,23]. To test this hypothesis, we examined the developmental genetic pathways that are enriched in the exaggerated legs through analyzing gene ontology of leg-biased genes (S3 Fig; S3 Table). Genes that are up-regulated in the third legs of males (in comparison with the first legs of males) were enriched in metabolic processes, transmembrane transport, and growth signaling pathways. Genes that are up-regulated in the third legs of females (in comparison with first legs of females) were enriched in signaling and metabolic pathways as well as response to stimuli and cell communication processes. Among the genes and pathways that were hypothesized to play a role in the regulation of growth-related exaggerated traits [3], many were identified in our datasets as differentially expressed between legs within the same sex or between the sexes (S4 Table; see [24] for sex-biased genes).

To test the role of these genes in trait exaggeration, we conducted an RNAi screen targeting about 30 candidates representing transcripts that were either leg-biased, sex-biased (up-regulated in male's or female's third legs), broadly expressed, or tissue-specific (S5 Table). This functional screen identified 2 genes, BMP11 (also known as *GDF11*) and *Ultrabithorax* (*Ubx*), the depletion of which unambiguously and reproducibly resulted in altered scaling relationships in the males. Some of the other genes tested were lethal or did not produce any detectable effect, thus preventing conclusive analyses of scaling relationships (S5 Table). Ubx is a tissue-specific Hox protein known to be confined to the second and third thoracic segments in water striders [25–28]. In our comparative transcriptomics dataset, Ubx is absent from the first legs, lowly expressed in the second legs, and highly expressed in the third legs of both sexes (S5A Fig), thus confirming its tissue-specific expression in *M. longipes*.

BMP11 is a known secreted growth regulator with a systemic effect in vertebrates [29,30]; however, its expression in waters striders is unknown. A phylogenetic analysis of the *BMP* gene family clustered this sequence from many water striders with *BMP11* of humans and mice, thus confirming their homology (S6 Fig; S1 and S2 Data). Our comparative transcriptomics analysis showed that *BMP11* is expressed in all legs and its levels of expression matched the differences in leg length along the body axis of both males and females (S5B Fig). A quantitative reverse transcription polymerase chain reaction analysis confirmed that *BMP11* is up-regulated in the third legs compared to the first legs of both males and females (two-way ANOVA; $p < 0.0001$; S5C Fig). Unlike comparative transcriptomics, however, qPCR revealed that *BMP11* is significantly up-regulated in the third legs of males compared to females (two-way ANOVA; $p < 0.0001$; S5C Fig). In addition, qPCR analysis revealed that *BMP11* is

expressed in the body of both sexes but with significantly higher levels in the females (two-way ANOVA; $p < 0.0005$; S5C Fig). This analysis confirms that *BMP11* is broadly expressed in *M. longipes*.

*Ubx* RNAi knockdown during nymphal development resulted in a 16% average reduction in males' third leg length (t-test: t = −3.5135, df = 62.904, *p*-value = 0.0008; S6 Table). However, this effect of *Ubx* RNAi was uniform across the range of male body size variation such that the intercept was shifted but the slope remained unaffected (Fig 2C; S6 Table for statistics). We also detected a small but significant effect in the second legs, and no effect in the first legs, consistent with the expression of *Ubx* in these tissues (Fig 2A and 2B; S6 Table). We did not detect any effect on average body length in males resulting from *Ubx* knockdown (S6 Table). *Ubx* knockdown in females also altered the intercept for the third legs but to a lower extent than in males despite the similar expression of *Ubx* in females (Fig 2C; S6 Table). This result first demonstrates that, in *M. longipes*, changes in leg length can be genetically decoupled from changes in body length, consistent with previous findings for other exaggerated sexually selected traits [4]. Second, this tissue-specific gene is associated with a change in intercept (i.e., increases the relative size of the third and second legs in a proportionate manner across individuals) but not in slope.

RNAi knockdown of *BMP11* during nymphal development affected the growth of multiple traits, consistent with the broad expression of this gene (Fig 3). *M. longipes* females have higher expression of *BMP11* in the body (S5C Fig) and larger mean body length (S6 Table). Depletion of *BMP11* RNA decreased the growth of the pronotum and significantly reduced mean body length by 8% in males and 12% in females (Fig 3A, 3B, 3G and 3H; S6 Table). Importantly, RNAi against *BMP11* resulted in a significantly reduced slope of male third legs from 3.1 to 2.0 (ANOVA, F-value = 9.6327, *p*-value = 0.002468) (Fig 3M; S6 Table). The slope for the second legs of males, which are also exaggerated, was significantly lower in individuals treated with *BMP11* RNAi compared to the controls (S7 Fig; S6 Table). Because small males generated through artificial selection or through poor diet treatment maintain the same allometric coefficient as large males [19], we conclude that BMP11 is necessary to increase the allometric coefficient of the third and second legs as well as mean body size in the males of *M. longipes*. Just like in males, *BMP11* knockdown reduced mean body size in females (Fig 3M; S6 Table). Unlike males, however, *BMP11* knockdown in females did not affect either slope or intercept, despite the up-regulation of *BMP11* in females' third legs compared to the 2 other legs (Fig 3M; S5 Fig; S6 Table). Altogether, these results indicate that BMP11 is a key growth regulator in *M. longipes*, playing a role in leg allometry (i.e., increases the relative size of the third and second legs in a disproportionate manner relative to the body) in males and mean body size in both males and females during development.

## Evolution of BMP11's role in regulating scaling relationships across *Microvelia*

Trait exaggeration, whereby second and third legs grow in disproportion to the growth of the body in males, is a derived state that separates *M. longipes* from other *Microvelia* spp. [19,31]. We therefore sought to determine whether the role of BMP11 in regulating scaling relationships is also a derived state in the lineage leading to *M. longipes*. To test this hypothesis, we examined the effect of *BMP11* RNAi on 2 additional sister species, namely, *M. pulchella*, a sister species of *M. longipes* and *M.* sp. (a species in the process of being described and for which we have not chosen a name yet) which is sister to *M. longipes*/*M. pulchella* [19]. Just like in *M. longipes*, *BMP11* RNAi also reduced the growth of the pronotum in both sexes of *M.* sp. (Fig 3 compare C to D; S6 Table). In *M. pulchella*, however, the pronotum does not cover the

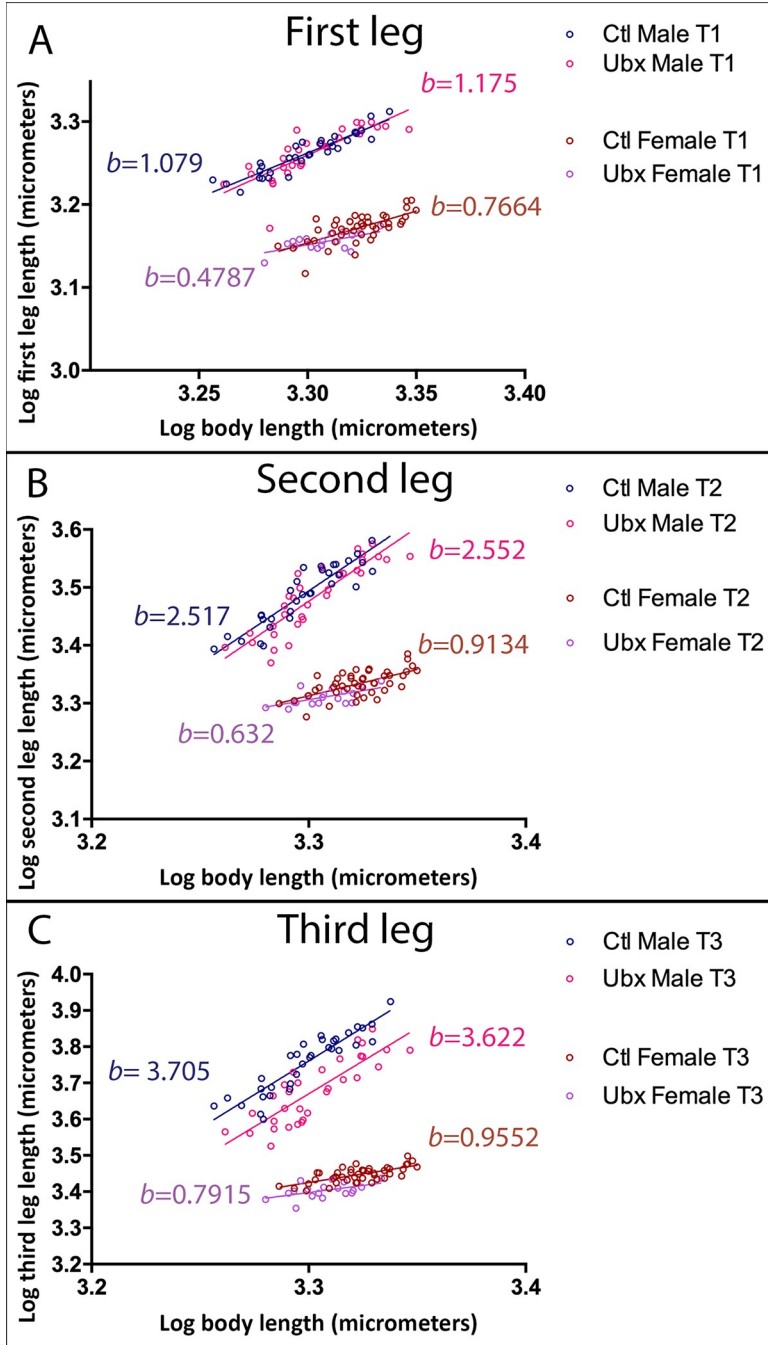

**Fig 2. Effect of *Ubx* RNAi on scaling relationships between leg length and body length in *M. longipes*. (A)** Scaling relationships are not altered in males or females by *Ubx* RNAi. **(B)** There is a subtle decrease in the intercept of second legs upon *Ubx* RNAi. **(C)** *Ubx* RNAi causes a significant decrease in intercept of the third legs in males. Note that body size is not affected by *Ubx* RNAi. Linear regressions were fitted to the log-transformed datasets. Statistical analyses were also performed on log-transformed data and can be found in S6 Table. Sample sizes are as follows: control males $n = 33$; control females $n = 44$; *Ubx* RNAi males $n = 32$; *Ubx* RNAi females $n = 18$. The data underlying this figure may be found in S6 and S7 Tables. Ctl, xxxx; RNAi, RNA interference; *Ubx*, *Ultrabithorax*.

mesonotum, and *BMP11* RNAi did not cause any detectable reduction in the growth of this trait (Fig 3E and 3F). In addition, *M.* sp. is the only species in this sample where males' third legs possess a prominent set of spines that are not found in the females, indicative of sexually antagonistic selection [32,33] (Fig 3I). Surprisingly, *BMP11* RNAi reduced or removed these spines (Fig 3J), indicating that this gene could also play a role in the development of discrete sexually dimorphic traits. Analyses of scaling relationships revealed that *BMP11* RNAi did not affect slope either in *M. pulchella* or *M.* sp. (Fig 3N and 3O; S7 Table; S7C, S7D, S7E, and S7F Fig). However, *BMP11* RNAi significantly reduced mean body size in both sexes of *M.* sp. and *M. pulchella* (Fig 3N and 3O; S6 Table). The size of the body and that of all appendages was reduced by *BMP11* RNAi in comparable proportions between males and females (S6 Table). We conclude that the role of BMP11 in increasing mean body size is ancestral and that its role in increasing allometric coefficient is derived in *M. longipes*. In addition to its role in regulating quantitative sexual characters such as hyperallometric scaling relationships, BMP11 also plays a role in the development of discrete male-specific traits such as the spines found on the third legs of *M.* sp.

## Link between BMP11 and fighting behavior in *M. longipes* males

We have shown that the evolution of exaggerated third legs, which are used as weapons by *M. longipes* males, was accompanied with a drastic increase in male fighting frequency to over an order of magnitude more compared to its sister species *M. pulchella* [19]. We have already shown that leg allometry and body size were genetically correlated through the role of BMP11 during development. To test whether the evolutionary concurrence between the exaggerated weapon and the increased male fighting behavior in *M. longipes* could also be the result of genetic correlation, we compared the frequency of fights between controls and individuals treated with *BMP11* RNAi in male–male competition assays (see Material and methods; S1 and S2 Videos). Similar to the controls, *BMP11* RNAi-treated males stood on egg-laying sites and signaled through ripples to attract females (S1 and S2 Videos; [19]). However, unlike the controls, these *BMP11* RNAi-treated males avoided fighting when other males approached egg-laying sites. Instead, all these males gathered around the female and tried to copulate instead of engaging in fights to chase rival males away (S2 Video). Sometimes, the males abandoned the site altogether without a fight. We quantified this "docile" behavior by calculating the frequency of fights on egg-laying sites and found that *BMP11* RNAi-treated males fought on average 12 times less than control males (Fig 4A; GLM: F-value = 29.96, df = 6, *p*-value = 0.0028). *BMP11* RNAi-treated females were attracted by male's signals on floaters, but we failed to observe any mating events during our trials (S2 Video). By contrast to *BMP11* RNAi- treated males, *Ubx* RNAi-treated males retained aggressive fighting behavior (Fig 4A), although the fight between 2 rival males lasted significantly longer compared to controls (Fig 4B; S3 Video). These results suggest that the disruption of scaling relationships of the exaggerated weapon, through RNAi, also results in the disruption of the fighting behavior associated with it. Therefore, BMP11, but not Ubx, is necessary to increase fighting frequency in *M. longipes* males. Collectively, our results show that in *M. longipes* males, BMP11 regulates the development of the exaggerated weapon and the associated fighting behavior.

## Discussion

We have shown that leg exaggeration in *M. longipes* males is associated with a specific signature of gene expression. Among the genes up-regulated in the exaggerated trait, we demonstrated that a broadly expressed growth factor, BMP11, increases allometric slope, whereas a tissue-specific factor, Ubx, increases allometric intercept. Finally, we have shown that BMP11 regulates the growth of the body, the disproportionate growth of the legs, and also increases aggressiveness in males. This indicates that these 3 male traits are genetically correlated

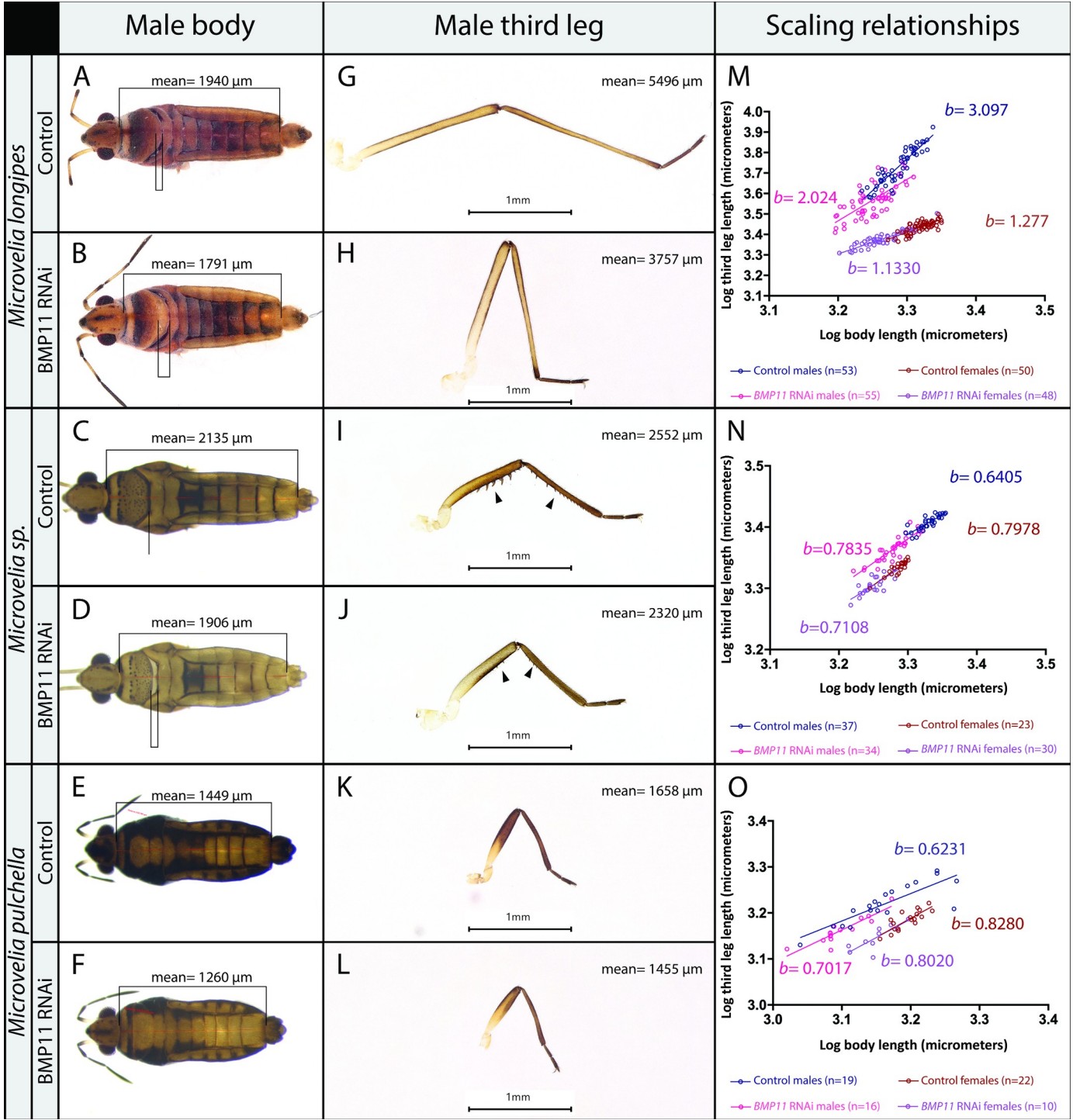

**Fig 3. Effect of BMP11 RNAi on the growth and scaling relationships of various traits in 3 species of *Microvelia*.** The pronotum in **(A)** *M. longipes* and (C) *M. sp.*, but not in (E) *M. pulchella*, covers the entire mesonotum (brackets in A and C). *BMP11* RNAi characteristically reduces the size of the pronotum in *M. longipes* and *M. sp.* (brackets in B and D). (**E, F**) This effect is absent in *M. pulchella*, consistent with the absence of this trait. (**G–L**) Only *M. sp.* males have a set of spines in the third legs (arrowheads in I), which are removed or reduced due to BMP11 RNAi (arrowheads in J). The lengths of the third legs are reduced in adult males and females. Statistical analyses can be found in S6 Table. *M. pul* stands for *M. pulchella*. (**M–O**) Effect of BMP11 RNAi on adult male and female scaling relationships between leg length and body length of *M. longipes* (M), *M. sp.* (N), and *M. pulchella* (O). Statistical analyses can be found in S6 Table. Comparisons of adult static allometries for each species between controls and BMP11 RNAi for males and females are shown in the keys below graphs. Linear regressions were fitted to the log-transformed datasets. Samples sizes for each species are indicated in the panels. Statistical analyses were also performed on log-transformed data and can be found in S6 Table. The data underlying this figure may be found in S6 and S7 Tables. BMP11, Bone Morphogenetic Protein 11; RNAi, RNA interference.

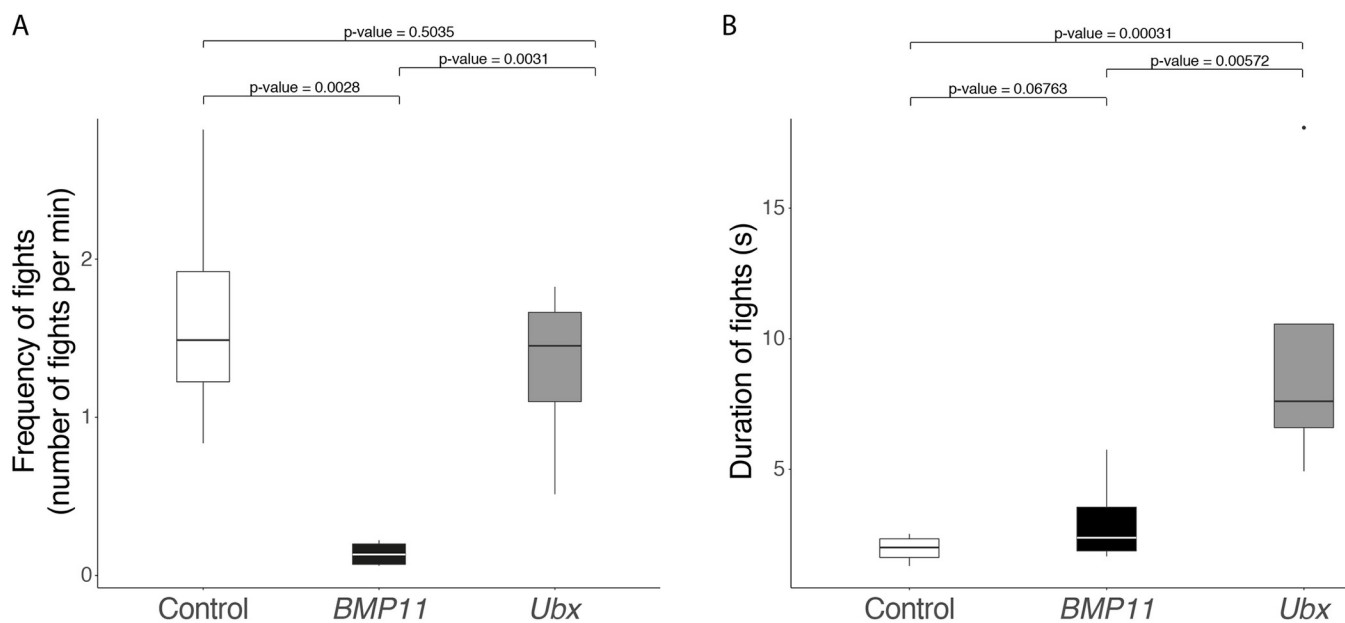

**Fig 4. Effects of *BMP11* and *Ubx* RNAi on male fighting behavior. (A)** Effect of *BMP11* and *Ubx* knockdowns on male fighting frequency (number of fights per unit of time) compared to control individuals. **(B)** Effect of BMP11 and *Ubx* knockdowns on male fighting duration compared to control individuals. The data underlying this figure may be found in S8 and S9 Tables. *BMP11, Bone Morphogenetic Protein 11*; RNAi, RNA interference; *Ubx, Ultrabithorax*.

through the pleiotropic effect of BMP11. How BMP11 changes fighting behavior in *M. longipes* males is unknown. A possible explanation is that the docile behavior of *BMP11* RNAi-treated males can be due to an adaptive response, whereby males recognize their small body size and change reproductive strategy. Another possible explanation could involve a role of BMP11 in the development of nervous system components associated with the fighting behavior.

Exaggerated traits represent striking cases of differential growth between various organs, and we are just beginning to understand the underlying developmental genetic processes [23,34]. Studies in various insects suggest that traits would achieve exaggeration via their increased sensitivity to systemic growth factors such as Juvenile Hormone or Insulin [4,7,8,35,36]. Our data also show that exaggeration, as seen in the second and third legs of *M. longipes* males, can be induced by growth regulating molecules other than hormones, such is the case for BMP11. These molecules are produced in excess by the exaggerated tissue, indicating an alternative path to trait exaggeration where overproduction of the growth factor occurs locally rather than through heightened sensitivity to a systemic factor. Whether and how this increased tissue-specific expression of growth factors is connected to systemic growth pathways remains to be tested [8].

Our experiments describe a common effect of BMP11 on mean body length in both sexes. By contrast to males, the effect of BMP11 on body length is decoupled from its effect on leg length in females. It is possible that increased body size has been favored in both sexes, through increased competitiveness in males and increased fertility in females [37,38]. Another explanation could be that smaller females were disfavored due to reproductive incompatibility as body size increased in males. On the other hand, the lack of correlation in leg length between the sexes may be a consequence of sexual conflict resolution through dimorphism [32].

We have previously shown that closely related members of the *Microvelia* genus differ in their slope, intercept, mean body length, and the extent of variation of leg length and body length [19]. Among the over 200 known *Microvelia*, *M. longipes* is the only species that combines large mean body length and high variation in both body and leg length. This suggests that the size of the legs and the size of the body can evolve the ability to grow independently or

in tight scaling relationship. Our data show that tissue-specific developmental regulators, such as Ubx, regulate trait growth independently of body growth, resulting in a change in intercept. BMP11 on the other hand, which is expressed in the legs and in the body, is necessary for systemic growth. Interestingly, our RNAi experiments show that the role of BMP11 in regulating the growth of the body is a conserved feature in this sample of species and this role does not seem to be sex-specific. However, the role of BMP11 in increasing slope is a derived state in *M. longipes*. This indicates that the input provided by developmental genes, likely in the context of their interactions within genetic networks, can evolve and result in various phenotypic outcomes, from discrete sex-specific phenotypes (as is the case of male spines in *Microvelia*. sp.) to quantitative variable traits such as body size and leg length.

Studies of scaling relationships have reported a strong stasis of the allometric coefficient at the microevolutionary time scale [11–13]. Even at the macroscale, slope is known to evolve slower compared to other scaling parameters such as intercept [11,12]. Several hypotheses were formulated to explain such evolutionary stasis, including a lack of genetic variation, strong stabilizing selection acting on slope, or a constraining effect of pleiotropy [11–14]. Our results, implicating BMP11 in regulating growth of body size and all 3 legs, lend support to pleiotropy as a major driver of evolutionary stasis in allometric coefficient. In contrast, Ubx, which regulates the size of the third legs but not body size, has a role in changing the intercept. This suggests that the genetic architecture of slope may have more pleiotropic effects than that of intercept, possibly explaining the observed differences in evolutionary stasis between these 2 scaling parameters. Whether this conclusion can be generalized to other traits requires additional comparative studies between species that have evolved different scaling relationships.

An important question is how trait exaggeration evolves despite the evolutionary stasis of allometric slope. We have shown that the evolution of hyperallometry in *M. longipes* is associated with increased mean body size and male aggressiveness compared to its sister species *M. pulchella* [19]. These changes are also associated with strong selection on males due to intense competition for access to females [19]. In this context, selection is expected to favor traits that increase male competitiveness. Strikingly, in *M. longipes*, BMP11 regulates mean body size, allometric slope, and male aggressiveness, all of which increase male competitiveness and likely fitness. It is possible that directional selection favored genotypes that regulate all 3 traits at the same time leading to the correlated evolution of these male traits in *M. longipes*. Pleiotropy would therefore become both a promoting factor for the evolution of exaggerated weapons and a constraining factor involved in the stasis of slope.

## Material and methods

### Establishment of growth curves

*M. longipes* is a hemimetabolous insect that develops through successive molts which represent exuvia of the previous nymphal instar. Using a population collected in *Crique Patate* near Cayenne in French Guyana, we raised each individual separately from first nymphal instar to adulthood. Throughout this process, each individual produced 5 exuvia corresponding to the 5 nymphal instars. We then collected the exuvia and the adult from each individual and measured their leg lengths (S1 Fig).

### Comparative transcriptomics: Experimental design and Principal Component Analyses (PCA)

Comparative transcriptomics was performed on RNA extracted from individual legs belonging to 2 inbred populations, raised through over 20 generations of brother–sister crosses in

laboratory conditions. During inbreeding, each of these 2 populations was selected either for long-legged (long-legged selected line) or short-legged males (short-legged selected line), respectively [19]. The sampling was as follows: 2 lines, 2 sexes, the 3 pairs of legs each replicated 3 times. Each replicate represents a pool of 20 individuals, randomly selected at the fifth nymphal instar. The raw Illumina reads were mapped to the genome of *M. longipes* [24].

We performed an analysis of gene expression variance of all genes expressed in the 3 pairs of legs. For this we selected genes with FPKM > 2 in at least 1 of the 3 legs. The filtering process and PCA analysis were performed in males and females separately (S3 and S4 Figs). For the PCA, we also corrected both for line and replicate effects that are intrinsic to our experimental design [24]. The latter effect matched the days where RNA was extracted from each sample. We corrected for both effects, using Within Class Analysis, with the R package 'ade4' version 1.7–15 [39].

## Identification of leg-biased genes

The number of reads per "gene" was used to identify differences in expression among the different legs using DESeq2, with the R package DESeq version 1.39.0 [40]. To identify the genes underlying the growth differences in the pairs of legs, we looked for genes that were differentially expressed between leg pairs (first versus third leg, first versus second leg, and third versus second leg) in each sex separately. All differential expression analyses were performed on the 2 lines combined as we aimed to identify genes involved in allometric slope, which is a common feature of both lines. We first filtered transcripts for which expression was lower than 2 FPKM in more than half of the samples after combining the 2 inbred populations and the 3 replicates per condition (2 lines × 2 legs × 3 replicates = 12 samples total). Transcripts with average expression that was lower than 2 FPKM in the 2 legs being compared were also discarded. Finally, we ran the differential expression analysis by taking into account the line and replicate effects. We called differentially expressed genes any gene with a fold-change > 1.5 and a Padj < 0.05.

## Gene Ontology analysis

Gene names and functions were annotated by sequence similarity against the NCBI "nonredundant" protein database using Blast2GO. The Blast2GO annotation was then provided to detect Gene Ontology terms enrichment (*p*-value < 0.05) using the default method of TopGO R package version 2.34.0.

## Nymphal RNAi

Double-stranded RNA (dsRNA) was produced for *BMP11* and *Ubx*. T7 PCR fragments of the 2 genes were amplified from complementary DNA (cDNA) template using forward and reverse primers both containing the T7 RNA polymerase promoter. The amplified fragments were purified using the QIAquick PCR purification kit (Qiagen, France) and used as a template to synthesize the dsRNA as described in [28]. The synthetized dsRNA was then purified using an RNeasy purification kit (Qiagen) and eluted in Spradling injection buffer [41] at a concentration of 6 μg/μl. For primer information, see S5 Table. Nymphal injections were performed in the line selected for long-legged males [19] at the fourth instar as described in [27]. In the first experiment, we injected nymphs in parallel with *Ubx* or *BMP11* dsRNA or with buffer as negative controls. For *Ubx* RNAi, we compared adults obtained from Ubx dsRNA injection to those obtained from buffer injection. For BMP11 RNAi, 2 additional experiments were added to ensure RNAi reproducibility and increase sample size. Those experiments were performed in parallel with nymphs isolated but not injected as negative controls. Nymphs

were placed in water tanks (22 cm long, 13 cm high, 10 cm wide), for which the water was changed every day, and fed ad libitum with crickets until they developed into adults. To control for RNAi specificity and efficiency, we used various methods as follows: (1) We performed quantitative reverse PCR on individuals injected with *BMP11* dsRNA and confirmed that *BMP11* mRNA levels were significantly lower compared to normal individuals (S8 Fig). (2) We used negative control individuals consisting in nymphs that were injected with buffer or noninjected (but reared in parallel to treatments). (3) We injected *M. longipes* with 3 different dsRNA preparations based on 3 different fragments of the BMP11 gene (two of which are non-overlapping) and obtained the same effect as assessed through measurements of pronotum size (S7 Table).

## Absolute quantitative reverse transcription polymerase chain reaction

This method was used to compare the expression of *BMP11* in the legs and body of males and females and to confirm the efficiency of RNAi knockdown. For *BMP11* expression, the 3 pairs of legs and the body from 30 fifth instar male and female nymphs were used, separately, to extract total RNA. To confirm the efficacy of RNAi treatment, 4 male and 4 female whole nymphs from both untreated and *BMP11* dsRNA-injected samples were used for total RNA extraction. After DNAse treatment, 400 nanograms (expression) or 1 microgram (RNAi) per sample were used for reverse transcription to produce cDNA. The qPCR was conducted using 2 *BMP11* primer pairs using cDNA from the legs and the body in 3 replicates. To determine the concentration of *BMP11* in each sample, we first determined the efficiency of 2 *BMP11* primer pairs to be 94.23% and 96.63%, respectively (see primer sequences in S5 Table). Second, we build a concentration curve using a *BMP11* PCR-amplified DNA template with incremental (factor of ×2) concentrations from 0.0488 femtograms to 100 femtograms. This curve yielded a slope of 3.66 and 3.60 and intercept of 14.9 and 14.5 for the first and second primer pairs, respectively. The concentrations of BMP11 in the various samples was calculated using the data from the curve. qPCR and RNA-sequencing data used to construct S5 and S8 Figs can be found in S10–S12 Tables.

## Behavioral assays

Male competition assays were performed using artificial puddles (containers 14 cm long, 10 cm large, and 4 cm high) containing 5 egg-laying floaters. Each replicate corresponded to a population of 5 wild-type females [19] and 10 males from a treatment (either 10 control or 10 *BMP11*-RNAi males). Analyses were performed on 4 replicates for each condition. Male and female interactions were recorded on a Nikon digital camera D7200 with an AF-S micro Nikkor 105 mm lens. Video acquisitions were taken a couple of hours after the bugs were transferred to the puddle. We defined a fight as an interaction between a focal male already present on a floater and a challenging male approaching the floater and inducing the 2 males to turn back to back and to engage into kicking with their third legs (S1 Video). If 2 contestants stopped fighting for more than 5 seconds, we counted the new interaction as a separate fight. For each replicate, video recordings last about 80 minutes (details about video record, number, and duration of fights are provided in S8 and S9 Tables).

## Statistical analyses

All statistical analyses were performed in RStudio 0.99.486. Comparisons of mean trait size and trait variance were performed on log-transformed data and used for scaling relationship comparisons. Winged individuals are rare and have different scaling relationships. We therefore excluded them from the analysis. We used standardized major axis (SMA) as well as linear

models (ANCOVA with body size as covariate) to assess differences in scaling relationships (using R package 'smatr' version 3.4–8 and ANOVA in R, respectively, [42,43]). Coefficients of variation were calculated using the R package 'goeveg' version 0.4.2.

To test for differences in fight frequency and fight duration, we used a generalized linear model with Gamma distribution, considering replicates as batch effects. Fights that lasted less than 1 second were standardized to 0.5 second.

Data deposited in the Dryad repository: https://datadryad.org/stash/share/2Q9tb7wD4R_iWkDSRAiOY2NsFuXCZE99o60C6Oeydk4 [43].

## Supporting information

**S1 Fig. Representative picture of the nymphal molts left by an individual during its post-embryonic development in *M. longipes* (here a male).** These molts were used to build a growth curve for each individual during nymphal development.
(TIF)

**S2 Fig. Covariation between male third leg length and duration of postembryonic development (nymphal development) until adulthood.** R-squared and *p*-value of the linear regression are indicated. The data underlying this figure may be found at https://datadryad.org/stash/share/2Q9tb7wD4R_iWkDSRAiOY2NsFuXCZE99o60C6Oeydk4.
(TIF)

**S3 Fig. Signature of trait exaggeration among leg-biased genes. (A)** PCA analysis of expressed genes in male and female legs separately. **(B)** MA plots of transcripts differentially expressed in the third and first legs in both sexes. Gray circles represent unbiased genes, while colored circles (blue in males, purple in females) represent genes significantly differentially expressed between the 2 pairs of legs at an adjusted *p*-value of <0.05. Venn diagrams illustrate the number of leg-biased genes (fold-change > 1.5) shared in females (purple) and males (blue), both for up-regulated genes in the third (top) and first legs (down). The data underlying this figure may be found at https://datadryad.org/stash/share/2Q9tb7wD4R_iWkDSRAiOY2NsFuXCZE99o60C6Oeydk4. MA, Log ratio and Mean average; NS, Non significant; PCA, Principal component analysis.
(TIF)

**S4 Fig. Boxplot showing differences in fold change (log2FoldChange) between males and females among the leg-biased genes.** The data underlying this figure may be found at https://datadryad.org/stash/share/2Q9tb7wD4R_iWkDSRAiOY2NsFuXCZE99o60C6Oeydk4.
(TIF)

**S5 Fig.** Levels of expression of *Ubx* and *BMP11* transcripts in *M. longipes* as revealed by comparative transcriptomics (A and B) and by quantitative RT-PCR (C). Note that for qPCR we also determined that BMP11 is significantly biased in female body (two-way ANOVA; $p < 0.0001$). The data underlying this figure may be found in S10 and S11 Tables. *BMP11, Bone Morphogenetic Protein 11*; RT-PCR, reverse transcription polymerase chain reaction; *Ubx*, *Ultrabithorax*.
(TIF)

**S6 Fig. Phylogeny of BMP family (protein and nucleotide sequences) comparing BMP sequences in multiples species of water striders with insects and vertebrates.** BF: *Brachymetra furra*, HH: *Husseyella halophyla*, ML: *Microvelia longipes*, MP: *Microvelia pulchella*, OB: *Oiovelia brasiliensis*, PB: *Platyvelia brachialis*, GB: *Gerris buenoi*, Pyr: *Pyrrhocoris apterus*, Tribolium: *Tribolium casteneum*, Droso: *Drosophila melanogaster*, H.sp1: *Hebrus sp1*, chicken:

*Gallus gallus*, human: *Homo sapiens*, mouse: *Mus musculus*. Nucleotide and protein alignment files can be found in S1 and S2 Data. The data underlying this figure may be found at https://datadryad.org/stash/share/2Q9tb7wD4R_iWkDSRAiOY2NsFuXCZE99o60C6Oeydk4.
(TIF)

**S7 Fig. Scaling relationships of the first and second legs in the males and females of *M. longipes*, *M.* sp. (From Cayenne, French Guiana) and *M. pulchella*.** There is no effect of BMP11 (GDF11) RNAi on slope in any of these tissues except the exaggerated second legs of *M. longipes* (**B**). The data underlying this figure may be found in S6 and S7 Tables. BMP11, Bone Morphogenetic Protein 11; Ctl, Control; GDF11, Growth Differentiation Factor 11; RNAi, RNA interference.
(TIF)

**S8 Fig. Quantitative RT-PCR confirms that BMP11 RNAi significantly reduces the levels of expression of BMP11 both in males and females (two-way ANOVA; p < 0.0001).** The data underlying this figure may be found in S12 Table. BMP11, Bone Morphogenetic Protein 11; RNAi, RNA interference; RT-PCR, reverse transcription polymerase chain reaction; WT, wild type.
(TIF)

**S1 Table. Leg measurements of males and females over the 5 nymphal instars and adult stage.**
(CSV)

**S2 Table. GLM statistics on ontogenetic leg allometries.**
(XLSX)

**S3 Table. Tables summarizing Gene Ontology terms for leg-biased genes across legs and sexes.**
(XLSX)

**S4 Table. Summary of genes/pathways known or suspected to be involved in regulating exaggerated trait growth.**
(XLSX)

**S5 Table. Identifiers, names, expression patterns, fold changes, and phenotypes of the genes screened by RNAi and primer sequences.**
(XLSX)

**S6 Table. Summary statistics of *BMP11* and *Ubx* knockdown effects in leg lengths, body length, and static allometries across the 3 *Microvelia* species.**
(XLSX)

**S7 Table. Raw measurements of control and RNAi individuals used in this study.**
(XLSX)

**S8 Table. Summary table of the competition assays between *M. longipes* controls and knockdowns.**
(CSV)

**S9 Table. Raw data number and duration of fights between *M. longipes* controls and knockdowns.**
(CSV)

**S10 Table. Results of qPCR in *M. longipes* male and female legs and body.**
(CSV)

**S11 Table. Counts of *BMP11* and *Ubx* expression using RNA-seq data in *M. longipes* male and female legs.**
(XLSX)

**S12 Table. qPCR test results of the efficiency of BMP11 knockdown using RNAi.**
(XLSX)

**S1 Video. Representative video of the fighting behavior in *M. longipes* control individuals.**
(MOV)

**S2 Video. Representative video of the fighting behavior in *M. longipes* BMP11 individuals.**
(MOV)

**S3 Video. Representative video of the fighting behavior in *M. longipes* Ubx individuals.**
(MOV)

**S1 Data. BMP11 nucleotide alignment.**
(FASTA)

**S2 Data. BMP11 protein alignment.**
(FASTA)

## Acknowledgments

We thank Leticia Arias and Pascale Roux for help with measurements, Juliette Mendes for help with insect rearing, Felipe Moreira for help with fieldwork and species identification, Romina Retamal-Figueroa for technical assistance, Francois Leulier, Dali Ma for comments on the manuscript, and Gael Yvert, Kevin Parsons, and Roberto Arbore for discussions. We thank Lois Taulelle and Hervé Gilquin for providing access to computing resources in the Pôle Scientifique de Modélisation Numérique (PSMN) at the ENS Lyon.

## Author Contributions

**Conceptualization:** William Toubiana, Abderrahman Khila.

**Data curation:** William Toubiana, David Armisén.

**Formal analysis:** William Toubiana, David Armisén, Abderrahman Khila.

**Funding acquisition:** William Toubiana, Abderrahman Khila.

**Investigation:** Abderrahman Khila.

**Methodology:** William Toubiana, David Armisén, Séverine Viala, Amélie Decaras, Abderrahman Khila.

**Project administration:** William Toubiana, Abderrahman Khila.

**Resources:** William Toubiana, Abderrahman Khila.

**Supervision:** Abderrahman Khila.

**Validation:** William Toubiana, Abderrahman Khila.

**Visualization:** William Toubiana, Abderrahman Khila.

**Writing – original draft:** William Toubiana, Abderrahman Khila.

**Writing – review & editing:** William Toubiana, David Armisén, Séverine Viala, Amélie Dec-
aras, Abderrahman Khila.

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
