## [Editor Report · Decision Letter 0]

3 Apr 2020

Dear Abdou, 

Thank you for submitting your manuscripts "Pleiotropy promotes male exaggerated weapon and its associated fighting behaviour in a water strider" and "Impact of male trait exaggeration on sex-biased gene expression and genome architecture in a water strider" for consideration as Research Articles by PLOS Biology.

Your manuscripts have now been evaluated by the PLOS Biology editorial staff as well as by an academic editor with relevant expertise and I am writing to let you know that we would like to send your submission - "Pleiotropy promotes male exaggerated weapon and its associated fighting behaviour in a water strider" - out for external peer review as a Short Reports paper. Our Academic Editor thinks, and we agree, that the other submission - "Impact of male trait exaggeration on sex-biased gene expression and genome architecture in a water strider" - does not possess the conceptual advances and sufficient mechanistic depth which are required for consideration at PLOS Biology.

Before we can send your manuscript to reviewers, we need you to complete your submission by providing the metadata that is required for full assessment. To this end, please login to Editorial Manager where you will find the paper in the 'Submissions Needing Revisions' folder on your homepage. Please click 'Revise Submission' from the Action Links and complete all additional questions in the submission questionnaire.

Please re-submit your manuscript within two working days, i.e. by Apr 05 2020 11:59PM.

Kind regards,

Di

PLOS Biology

---

## [Decision Letter · Decision Letter 1]

12 May 2020

Dear Dr Khila,

Thank you very much for submitting your manuscript "Pleiotropy promotes male exaggerated weapon and its associated fighting behaviour in a water strider" for consideration as a Short Reports at PLOS Biology. Your manuscript has been evaluated by the PLOS Biology editors, an Academic Editor with relevant expertise, and by three independent reviewers.

In light of the reviews (below), we will welcome re-submission of a much-revised version that will address all of the reviewers' concerns. You will need to comprehensively modify the manuscript in a way that it will stand alone, and you can reference your genome study which is deposited in BioRxiv. We cannot make any decision about publication until we have seen the revised manuscript and your response to the reviewers' comments. Your revised manuscript is also likely to be sent for further evaluation by the reviewers.

We expect to receive your revised manuscript within 2 months. Please note given lab shutdown due to the COVID-19 pandemic, we are flexible regarding turnaround time for revision.

**IMPORTANT - SUBMITTING YOUR REVISION**

*Re-submission Checklist*

*Published Peer Review*

*PLOS Data Policy*

*Blot and Gel Data Policy*

Sincerely,

Di Jiang

PLOS Biology

REVIEWS:

Reviewer #1 (Arnaud Le Rouzic, signed): Review of "Pleiotropy promotes male exaggerated weapon and its associated fighting behaviour in a water strider" by Dr Toubiana and colleagues, PBIOLOGY-D-20-00810R1. 

This manuscript reports an interesting set of results on the genetic architecture of the leg length in a species of water strider. In this species, the third pair of legs is much longer in males than in other related species, which is strongly suggestive of sexual selection. The authors show that (i) the leg length difference is due to a change in the limb development at the end of the nymphal stage, (ii) a set of genes which expression is associated with the third legs length could be identified, (iii) two genes, BMP11 and Ubx, could be validated by RNAi knockdown; the knockdown had a strong effect in allometric slope for BMP11 but not for Ubx; (iv) RNAi genotypes were characterised by a different fighting behaviour. The authors conclude that this interaction pattern between body size, leg length, and pleiotropy could explain the emergence of this specific, sexually selected trait. 

I am rather balanced about the manuscript. In the one hand, I found the result on the genetic architecture of allometric slopes very interesting. Whether or not allometry participates to the selection response (and even whether or not it is evolvable at all) remains an open question, and the results presented in the paper suggest that allometry is indeed evolvable, and that in addition allometric slopes and intercepts might be affected by different genes. As far as I can tell, this genuinely improves our understanding of the genetic bases of allometric relationships. The evidence based on RNAi "only" (not underestimating the complexity of genetic studies in non-model organisms) might need to be confirmed independently, but this seems to match perfectly the scope of Plos Biology short reports ("Short Reports are research articles that may be preliminary, based on a small number of experiments that might not completely flesh out the biological phenomenon under study"). On the other hand, the manuscript suffers from numerous weaknesses and inaccuracies, some being benign, others being more embarrassing, up to make peer-review uncomfortable due to the lack of information (or to the lack of consistency between different sources of information). 

*** Major concerns ***

1. I have the feeling that the manuscript may have been deeply re-organised several times. This is not a problem per se, but the reader often lacks basic information to understand the authors' reasoning. Moreover, the manuscript abuses references to a companion paper, which was provided as supplementary material. Yet, this reference should not preclude self-consistency of the current manuscript, and I was rather confused about the lack of basic information about the experimental protocol (how many individuals measured? What is the biological material (population, inbred lines, wild-caught vs lab-raised individuals? There are a couple of references to selected lines from ref 19, line 270 mentions "the two inbred populations" (?), this was genuinely confusing). How many populations / samples / replicates for the transcriptome analysis? Line 295 refers to "controls" (what are they?). Even more embarrassingly, I could not find any of the supplementary tables (in the supplementary PDF, there are 11 sup figures, and the captions of the supplementary tables and movies; I had a link to the movies but not to the supplementary tables), and the figure resolution was so low in the main document that I could barely read the captions. When digging into the companion paper or counting the data points in the figures, sample sizes and replication conditions seem adequate in most cases, so I don't think that the authors are hiding details on purpose; I assume that this rather comes from copy-paste etc. across several papers without checking that the necessary information was provided. Paradoxically, the manuscript gave a fair amount of unnecessary details (such as the focal of the camera for the behaviour movies) without making the experiment reproducible (how large were the artificial puddles, what distance was considered to define that the fight occurred "near a floater", how long were the recordings?). Note that the statistics are not reproducible (I did not find any link to the code, and key analyses (such as the power-law fit) was not documented beyond the fact that the authors used RStudio). 

2. Allometry is the central topic of the paper (and I think the title should be changed, see below). Yet, the introduction does not introduce basic facts about allometry; the reader is thus expected to be familiar with the manipulation of allometric slopes and intercepts, which may not match the requirements of a wide-audience journal. In addition, some sentences appear to be very confusing, e.g. line 51, "a genetic link between [...] traits and other sexual traits such as male body size": isn't it exactly what allometry is about? , or line 62, "the evolution of hyperallometry may be linked to the evolution in increased mean [...] body size": does this make sense?. To be honest, I thought at that point that the authors were simply confused about what allometry is about; Details in the methods and in some figures made me changed my mind; at least, allometry was probably measured properly, and discussed in a reasonable way in some parts of the manuscript. Yet, the authors made the strange choice of working on the trait scale and fit non-linear (power law) model, which makes everything more complicated and more difficult to interpret. Why not running the whole analysis on log traits? This would (partly) fix the heteroscedasticity that is obvious from figure 3A and 3C, make allometric relatioships linear, make it possible to use a traditional linear models ( ~ sex + leg + genotype + interactions) to compare allometric slopes and/or intercepts with t-tests... If necessary, the figure axes could be log axes so that the original meaning of the trait measures is not lost. At least, working on a log scale would have avoided what is probably an interpretation error of fig 1B, as the authors observe "a dramatic increase in leg length [...] between the 5th instar and adult males" (line 81) (from the figure: from about 3000 to 6000 µm). However, the length at the 4th instar seems to be about 1500 µm, which would mean that the leg length is simply doubling betweeen larval stages, including the last transition, and that there is nothing specific about this particular developmental stage. 

3. The manuscript reports three experimental results, which have different strengths. 

(i) The RNAi + length measurement experiment seems the most convincing, even if it probably lacks replication. (ii) The behavioral experiment is also convincing, although here again the possibility that the causal effect of RNAi is not really demonstrated. (iii) The transcriptomics analysis is very descriptive (and probably very noisy), and I am not sure it brings a lot of understanding to the main question. It might be that the only interesting result of the transcriptomics approach to the question raised in the introduction is to shortlist a set of genes that are overexpressed specifically in male third legs, but is remains unclear whether the list of genes from sup table 5 was derived from the transcriptomics approach. To be fair, reporting transcriptomics results that does not seem to be informative nor reliable is the norm in the field, but I found the ratio explanatory power/effort to understand fig 2 particularly disappointing.

Yet, none of them provide formal support to the conclusion from the manuscript title "pleiotropy promotes male exaggerated weapon"; this is rather a reasonable hypothesis that can be derived from the results. Note that the story of BMP11 being an example of a "magic" gene that may influence several traits that all need to evolve simultaneously under sexual selection might very well be true, but the results presented in this paper do not prove that it is the case (and even less that BMP11 actually evolved under sexual selection in this species). 

I am a bit concerned that by trying to juxtapose results of different strengths (especially by including transcriptomics results that are difficult to interpret in the frame of the evolution of allometry) to address a more ambitious question, the authors may have weakened the paper. 

4. Some methodological details of the paper make me uncomfortable

4.1. The exclusion of putative "unaffected RNAi" individuals from phenotypic measurements (line 299) is dubious. Sup fig 10 clearly shows that pronotom/mesonotum ratios are not so good at discriminating, and no indication about the correlation between this ratio and body size is provided. Validating the exclusion by checking that excluded phenotypes behaves as control individuals looks rather circular, and I think the authors should consider measuring more accurately the size of the bias induced by such a sorting procedure. Not knowing the number of removed individuals is a bit stressful here -- due to the lack of details, here again I will have to assume that this was reasonable. 

4.2. There is an issue with the behavioural results: most of the observed differences are not significant. However, when looking at figure 4, the distributions of observations are totally non-overlapping, which makes me think that the statistics are flawed, and that the authors' approach is deeply under-powered. My interpretation is that the authors have recorded four replicates of the behavioural experiment involving five females and ten males for each treatment. During each experiment, many fights have been recorded (if the cryptic "frequency" Y axis in fig 4A means "number of fights", then almost a thousand fights were recorded). But in the t-test for fight duration, it seems that the authors have just included the average duration of fights, as if there was only one fight in each experiment, and n=4 observations per replicate. I think that a much better statistical framework would be to run a generalised linear model with a Poisson family for the number of fights (fig 4A), which should increase the power of the test (no need to estimate the variance for count data, as the variance is expected to be the same as the mean); the duration of fights should be analysed based on individual fights, possibly considering the replicate as a random effect to account for possible batch effects. This should definitely display significant differences across treatments, and avoid highlighting a dubious difference between "non significant" and p = 0.08 (cf figure 4B).

4.3. I found the interpretation of transcriptomics results unclear and questionable. The authors found a convincing "legs" effect (fig 2A) in both sexes from the full expression pattern, but the fact that they propose two independent analyses for males and females suggests that the pattern might be confusing when both sexes are considered together (in this context, I don't understand how the authors could conclude that the first axis is similar across sexes line 101, but I might have misunderstood something). Then they isolate genes that are over or under-expressed in the third legs in both males and females, which (as almost always in transcriptomics studies) generate a lot of sets of genes with little overlap. Gene ontology returns different patterns in males and females, which seems rather odd, and the clustering illustrated in fig 2 does not seem to be very insightful (for instance, genes overexpressed in female third legs display the most similar pattern when comparing male third legs and female first legs...). Some conclusions from the text (e.g. line 122, "the third legs express a considerable set of common genes between males and females") are vague (what is "considerable"?) and not supported by the figures (is it really surprising that the same genes are expressed in the legs of both sexes anyway?). 

*** Details ***

* Line 61, "large body size and high variation": in the only other species that is analysed (e.g. black in fig 3), the body size variation seems even larger than in the focal species. The variation is legs size could simply be due to a change in allometry. 

* Line 67, and other places in the ms: the link between exaggerated weapon size and allometry is far from clear. Disproportionate organ size can either be due to a change in allometry and/or a change in size without a change in the allometric coefficient (if > 1). 

* Line 105: "the third legs diverged more form the first legs in males than in females": this looks in contradiction to a previous statement line 98, "we failed to detect any correlation between such variation in morphology and variation in gene expression". I am not sure to understand why selecting a set of genes differentially expressed between legs 1 and 3 generates a pattern that was not present in the full set of genes (is it just a problem of statistical power?). 

* Line 114, 63% of leg-biased genes were common between the sexes (fig 2C): fig 2C indicates 92+157+232=481 genes overexpressed in the first legs vs third legs, 157 of them (32%) being common between sexes. I am not sure to understand what were the reported 63%.

* Line 126, "extravagant signals [...] are predicted to occur in good condition individuals": how does this observation combine with the allometric argument? If high fitness males are also larger, then it is perfectly compatible with the fact that there is positive allometry involved (in which case the fitness argument is not really central, since body and organ sizes are constrained by the allometric relationship, and the association large weapon <-> high fitness is unavoidable). If this argument ignores allometry, then it means that organ size is (at least partly) independent from body size, and this suggests that the main question about allometry is not relevant (allometry becomes a statistical association which does not derive from biological constraints). 

* Line 144: I am not sure to understand why BMP11 is defined relative to its vertebrate homolog (including in the full phylogeny of Sup fig 7), there is no ortholog in insects (including D. melanogaster?). 

* Line 168 and 192: good practices in statistics require to report full p-values (and not only p < 0.05, which makes meta-analyses more complicated). 

* Line 209: this is an interesting (but confusing) use of "plastic response", as no environmental response is involved here. 

* Line 255: there is no fig 5C.

* Line 240: I am confused by the claim that trait exaggeration evolves despite the evolutionary stasis of slope. This is clearly not the case here. 

* Line 262, "analysis of variance", do you mean "principal component analysis"?

* Lines 350, 361: in the references, some author names are not capitalised. 

* Figure 2, I found the Venn diagrams rather confusing (the overlapping part is not proportional, and the diagrams scale is different in fig B and C. All diagrams seem to be scaled relative to the female count, which does not seem very intuitive). The color scale (low <-> high) is particularly uninformative. 

* Fig 3: state that Mpul stands for M. pulchella in the caption of figs 3A and 3B. It would be great to have the colour code indicated in the figure (perhaps replacing the R^2, which is not informative in a figure since it is easy to eyeball the fit?). 

* Fig 4A: "frequency of fights" is unclear (is it "number of fights"?). 

Signed: Arnaud Le Rouzic

Reviewer #2: This manuscript was coreviewed by a PI and a postdoc.

Toubiana et al. provide here a very interesting study that combines descriptive, observational data about gene expression with experimental, functional data about the developmental roles of important genes. Their results show a very satisfying connection between the development of an exaggerated sexually dimorphic trait and on male behavior, indicating a pleiotropic role for one of the genes studied, BMP11. Although not yet understood mechanistically, this link between behavior and morphology is expected to have interesting evolutionary consequences. The authors have done a fine job tying their results to outstanding questions about the major hypotheses relating to the evolution of exaggerated weaponry. However, we had to struggle in multiple places to figure out what analyses were done and thus we believe that the manuscript would benefit from clarification of some key components prior to publication. So modified, this would make it an appropriate and very accessible article for PLoS Biology's broad audience. 

Our substantive concerns with the paper are primarily about places where the paper does not make clear what comparisons are being discussed and what exactly the underlying data are. These concerns likely stem in part from the short format of the manuscript, in which methods have been very condensed.

Data availability: 

Please ensure that the bioproject PRJNA610161 is the correct accession number and that the project is made public by the time of publication. As of 4/27/20, this accession number does not resolve to a public project. (You can release the project without releasing the data.) 

Additionally, we did not have access to the supplemental tables and thus we were unable to check whether they included the data we would expect—i.e., tables that include the data necessary for reanalysis of both allometry and gene expression data. (A request was submitted to the PLoS Biology team on 4/24/20, and we received a response that the request has been forwarded, but we have not received the tables; we did get to enjoy the supplemental videos and figures.)

Aims/analyses/interpretations:

Phrasing of main aim: "determining the genetic architecture) (l. 54) and "determining the genetic basis (l. 69) both lead readers to expect an approach that identifies causal genetic variation, whereas the research approach used here identifies genes that are differentially expressed and functionally important, but may or may not be the loci responsible for the evolutionary changes (even in the context of these selected lines). Thus, we suggest rephrasing the aims.

Experimental design: The methods should briefly state the sampling scheme for the transcriptomic data: i.e., that the starting material was lines selected for larger differences in leg size, and that gene expression in 2 lines * 2 sexes * 3 replicates *3 legs were characterized, rather than refer to the companion manuscript for it..

PCA analysis (results on l.96-102): We question the interpretation of this analysis and recommend dropping it from the manuscript. If it is maintained, then a clearer explanation (including methods: e.g., what genes (all? leg-biased? different subsets in the 2 sexes?), how was PCA implemented) is needed both of what was done and of how the result supports the interpretation. We do not think it is meaningful to compare the % of variance explained by axis 1 between males and females if these are separate, independently run analyses (which is our impression). Even if total variation (not just %) were greater in one than the other, additional work would be needed to demonstrate that leg3 was the cause and further to demonstrate that a specific set of genes is involved (e.g., loading information).

Differential expression analyses (results on l. 103-123; methods on l. 267-275): 

It was difficult to figure out which comparisons were used to identify genes as 'leg-biased', even after having read the relevant sections of the companion paper. We concluded that the analyses were done on pairs of legs (1 vs. 3 and 1 vs. 2) from 2 lines * 3 replicates in each sex separately. If that's correct, then including sampling design in methods and changing l. 103-105 as follows would clarify:

"To identify the genes underlying the growth differences in serial homologs, we identified genes that were differentially expressed between leg pairs (first vs. third leg and first vs. second leg) in each sex separately."

(While the term 'leg-biased' is parallel to 'sex-biased', it may be more prone to being misunderstood as implying a comparison between legs and some other tissue. Thus, we wonder if this term could be avoided—but we didn't come up with a good alternative.)

We remain unclear on what the set of "upregulated genes restricted to male third legs" is; we could not match the possibilities we considered for the number of genes given (489) and were unclear what 'restricted' means here. Figure (2B) shows 273+56=329 genes in the Venn diagram up regulated in the male T3 leg. Thus, we would have expected the 489 to be either 329 + 56 or 329 or some smaller number if genes upregulated in other contexts (e.g., male T2) didn't count. Is this a typo, or are you referring to a different defined set of genes in the text versus the figure? Clarifying this should also help clarify what is meant by the statement on l. 109-111 that the male regulated genes are "similarly expressed" in females. (This hierarchical clustering result may be more or less forced by the way in which genes were chosen for analysis—i.e., the set of genes displays maximal differences between male T1 and T3 (and perhaps isn't DE between female legs? Despite the clustering pattern, Figure 2B (right) gives an overall impression that expression levels are not similar between female T1 and T3).

l. 114: where is the 63% figure coming from. Based on figure 2C, I would have expected the figure to be 157/(92+157+232) = 33%

Supp. Fig. 2 (and associated ANCOVA test on l. 83): We would expect that development time was potentially predictive of leg length, and thus that it should be on the x-axis (rather than the other way around). Also, if the ANCOVA was applied to this analysis, please make clear what the covariate was.

We encourage a supplemental methods section that covers in more detail the methods you're referencing from the Genomics companion paper in supplemental methods for this paper. Yes, it's duplicative, but if someone has access to one and not the other, they will thank you for the duplication. Additionally, sharing the analysis code would be helpful.

l. 265 (and elsewhere): Please include the R package name and the version number in the main methods text for all R packages used, as was done for pvclust (including also the version of R, which is separate from the version of RStudio).

Figure 2 comments: We recommend that in genes be sorted by expression values for the focal leg type. For example, if the male heatmap in 2B were sorted by expression of T3 leg (high > low) it would be much more obvious that the female T3 leg expression is very different from male T3 leg. 

Minor points/typos, etc.

1) Line 278: How many genes were included in this hierarchical clustering analysis? 

2) Line 300-301: Since you are discarding some data points (legitimately!) I recommend including the percentage of injected adults that had the short pronotum phenotype versus those that didn't for both controls and experimentals.

3) Lines 127-129 ("Proposed mechanisms for the development of exaggerated phenotypes include increased sensitivity or the production of growth factors by the exaggerated organ [4, 25].") That sounds to me like you're talking about two distinct hypotheses. But it seems like the next sentence is referring to a single hypothesis ('this hypothesis'). Please clarify—it may be as simple as changing 'this hypothesis' to 'these hypotheses'

4) Lines 150-166: To make this more accessible to a general bio audience -- In these sentences that sum up the results, can you rephrase what change in intercept and coefficient means in terms of relative and absolute sizes of legs and bodies? 

5) Line 156-158: Perhaps rephrase sentence as follows 'The allometric coefficients for the first and second legs of males were also lower than in the corresponding controls, but the changes were smaller than for the third legs.'

6) Line 164: Should be "….closer to those of its sister species…" 

7) Line 230: "Studies of scaling relationships have reported" rather than just "reported"

8) Figure 1 - Line 439 "Power low regression" should be power law regression.

9) Figure 4: - legend, the word "defects" should be "effects". 

Reviewer #3: Review of Toubiana et al

This is a really nice and very interesting paper describing the role the BMP and Ubx play in regulating the morphological scaling of the third leg in male water strider of the species, Microvelia longipes. Males of this species use their exaggerated third legs to compete for access to females. The authors used transcriptomics to look at differences in gene expression between the isometric first and hyperallometric third leg, in males and females. Using these data they constructed a list of 30 candidate genes that they functionally tested using RNAi. They found that two, BMP11 and Ubx had an effect on scaling. Specifically BMP11 reduced body size in males and females, and reduced the slope of the 3rd leg-body scaling relationship in males, while Ubx had no effect on body size, but reduced the intercept of the 3rd leg-body in males and females. At the same time knock-down of BMP11 reduced the aggressiveness of males.

The paper is significant in that it is one of the few that identifies a developmental mechanism that controls the slope and intercept of a morphological scaling relationship. I certainly think that it is of sufficient general interest to be published in PLoS Biology. Nevertheless, I have a number of suggestions about how the paper can be improved. The tables were not included in my review copy of the supplementary material, so some of my concerns may be addressed there. 

1) A key finding of the paper is that knock-down of BMP11 changes the slope of the 3rd leg-body scaling relationship. Looking at figure 3C, it is possible that the 3rd-leg-body scaling relationship of the BMP11-RNAi males may be an extension of the wild-type scaling relationship but in smaller males. That is, among small wild-type males, the slope of the 3rd-leg-body scaling relationship is shallower. If this were the case, then it is possible that BMP11.RNAi simply reduces body size, with no additional effect on scaling. To test this, the authors should look at scaling in males that are small, for example by reducing access to food during development. If the author's hypothesis were correct, then these small males should have the same body size as BMP11-RNAi males, but a steeper leg:body scaling relationship. These data would also confirm that the pronotum/mesonotum ratio does not increase with an increase in body size. If it does, then exclusion of BMP11-RNAi individuals with high pronotum/mesonotum ratios from the analysis of the effects on BMP11 on 3rd-leg-body scaling is not reasonable: the authors are simply excluding larger individuals with steeper leg:body scaling. 

2) In several places, the authors confound the necessity with sufficiency. For example: L158: "Therefore, BMP11 increases both the allometric coefficient of all legs and mean body size in the males", and L197: "These results suggest that BMP11, but not Ubx, increases fighting frequency in M. longipes males." Both these statements argue implicitly that BMP11 is sufficient to increase the allometric coefficient or fighting frequency. Since the authors have not increased BMP11 expression, but only decreased it, they can only be certain that BMP11 is necessary for the hyperallometry of the legs, and for increased fighting behavior, but not that it is sufficient. 

3) The authors do not provide evidence that BMP-RNAi or UBX-RNAi knocks down expression of both these genes. They should do so.

4) I did not have access to the table showing the statistics used to test the hypotheses described in the paper. They need to be included for a complete review. In particular, I am not sure how the authors tested for the effect of BMP-RNAi on variance (L150)

Minor points:

5) The section on transcriptomics of the different legs in males and females is difficult to follow. In particular, the authors do not appear to be using the data to test a clear hypothesis about how gene expression is regulated in the legs in males and females. What do these patterns of gene expression mean? would these patterns look like under different hypothetical mechanisms of leg exaggeration? What would you expect the pattern of PC1 and PC2 to look if 

morphological divergence in serial homologs between the sexes was the "result of global change in gene expression but rather from specific sets of genes" (L101)? The title is also misleading: "Leg exaggeration is associated with a specific signature of leg-biased gene expression". Given that only gene expression in the legs was measured, of course the gene expression is leg-biased. This section should be re-written or excluded. 

6) Including color and shape keys in the figures would help the reader interpret the charts.

7) For Figure 3, the authors should indicate what M.pul means. It took me a while to realize it meant M. pulchella.

---

## [Decision Letter · Decision Letter 2]

12 Feb 2021

Dear Abdou,

Thank you for submitting your revised Short Report entitled "The growth factor BMP11 is required for the development and evolution of a male exaggerated weapon and its associated fighting behavior" for publication in PLOS Biology. I've now obtained advice from two of the original reviewers and have discussed their comments with the Academic Editor. 

Based on the reviews, we will probably accept this manuscript for publication, provided you satisfactorily address the remaining points raised by the reviewers. Please also make sure to address the following data and other policy-related requests.

IMPORTANT:

a) Please change the title to include a mention of the organism studied; we suggest: "The growth factor BMP11 is required for the development and evolution of a male exaggerated weapon and associated fighting behavior in a water strider"

b) Please attend to the remaining requests from reviewers #1 and #2.

c) Please address my Data Policy requests (see further down).

We expect to receive your revised manuscript within two weeks. 

*Published Peer Review History*

*Early Version*

Best wishes,

Roli

Senior Editor,

rroberts@plos.org,

PLOS Biology

DATA POLICY:

Regardless of the method selected, please ensure that you provide the individual numerical values that underlie the summary data displayed in the following figure panels as they are essential for readers to assess your analysis and to reproduce it: Figs 1BC, 2ABC, 3MNO, 4AB, S2, S3AB, S4, S5ABC, S6 (alignment files), S7ABCDEF, S8. NOTE: the numerical data provided should include all replicates AND the way in which the plotted mean and errors were derived (it should not present only the mean/average values).

I note that some of these data may already be in some of the Supplementary Tables (e.g. Tables S7, S8, S9); if so, please clarify, and re-name these as Supplementary Data files (e.g. S1_Data, S2_Data...).

REVIEWERS' COMMENTS:

Reviewer #1:

[identifies himself as Arnaud Le Rouzic]

Review of "The growth factor BMP11 is required for the development and evolution of a male exaggerated weapon and its associated fighting behavior" by Dr Toubiana and colleagues, PLOS BIOL MS D-20-00810R2. 

This is a deeply revised version of MS D-20-00810R1 I had the opportunity to review last year. My opinion about the manuscript was balanced, as I found the topic and the results very interesting, but I had major concerns: 1) a lot of data and methodological information were missing, 2) it was not entirely clear that allometry was analyzed and interpreted properly, 3) transcriptomics results were difficult to interpret and not important for the paper, and 4) there were some methodological issues with some analyses. 

The authors have substantially rewritten the paper and clarified many issues, both in the ms and in their response to the reviewers. The revised version is quite convincing, I liked it a lot, and I have no further reason to oppose publication. 

My point #1 was completely addressed. The authors explained that two companion papers had to find different publication routes, and that the state of the previous version reflects an incomplete split. 

Point #2 has been largely addressed. Yet, I am still not completely convinced by the authors' interpretation of what happens between the 5th instar and the adult stage. It seems to me that the authors' narrative relies on the idea that "something" (a physiological discontinuity) happens there (the authors chose to extract RNA for transcriptomics at that time point). I totally agree that third legs grow substantially between the 5th instar and the adult stage, more than between any previous developmental stages. However, my interpretation of the growth curves (fig 1B) and supplementary tables 1 and 2 is that the development is possibly allometric (difficult to confirm without reanalyzing the data including body size at each larval stage). This is probably a matter of interpretation, but if the development of legs follow an allometric trajectory, nothing "dramatic" occurs at the 5th instar beyond the fact that growth curves diverge more and more on the arithmetic scale (a mere consequence of plotting growth, which is essentially a geometric process, on an arithmetic scale). On the opposite, if the allometric trend breaks down for the 3rd leg (i.e. leg length does not follow a power law), then I would agree that the pattern is discontinuous and that something special happens here. This, of course, should be regarded as a minor conceptual disagreement between the authors and a reviewer, and does not require a change in the manuscript. 

The focus on transcriptomics has been largely reduced, which fixes my point #3. 

Finally, the authors ran new RNAi injections and ran the analysis without excluding control-looking individuals (point #4.1), redesigned the statistical analysis of the behavioral assay (point #4.2), and fixed interpretation issues from the transcriptome analysis (point #4.3). 

*** Minor suggestions:

* Lines 244 and 245: it is probably necessary to indicate "allometric slope" and "allometric intercept". 

* Line 321: PCA and analysis of variance are two different analyses.

* Figures 2 and 3(M, N, O): the x and y ranges are much wider than the data range, making it difficult to read the figures. I am not sure it is justified to plot all data (across sexes, legs, and species) on the same x and y range, since direct comparison is not of interest here. 

Reviewer #2:

Overall, I found this revised manuscript substantially clearer and improved. It is much easier for readers to figure out what was done; the new functional data from two closely related species are an interesting addition; and the very strong effect of BMP11 on both behavior and the relationship between leg length and body size is unexpected and fascinating in the context of the evolution of sexually selected traits. 

There are, however, still a few places where I think the ms. would benefit from additional clarification or where more caution in interpretation may be called for:

l. 68-69: unclear why large body size is relevant. This isn't highlighted in the background nor is it inherent to consideration of allometry. (But it's a point that's returned to as a reason this species was chosen in several places.) The relevance of large body size either needs to be explained or statements about it should be removed. 

l. 259-260: Ubx was expressed at high levels in T3 legs long before the hyperallometry evolved. So I don't think this can be taken as evidence of an alternative pathway to trait exaggeration that's being argued for here. Additionally, I don't recall evidence that BMP11 is NOT upregulated in the sister species. This would need to be the case to support this central claim that it's overproduction of the growth factor that is causing increased leg growth. The data seem equally compatible with alternative interpretations about what the evolutionary change was, such as that it's a change in the expression of something downstream of BMP11 or in the responsiveness of something to BMP11 signaling. In the absence of comparative expression information, I think greater caution in interpreting the results is warranted.

Related to results interpretation, I was unclear what the relationship of the three studied species is. Among the three, if sp. nov. and pulchella are most closely related, with longipes sister to a clade that contains both of them, then the authors should be more cautious about interpretations of ancestral versus derived states (as there is not direct evidence about the polarity of a change when the difference distinguishes sister groups). One direction of change may be more plausible based on other considerations, but the distribution of traits on the tree would not be providing additional supporting evidence. (On the other hand, if one of the two additional species is more closely related to longipes, and the two additional species have the same trait value, this comment can be ignored, because the result would help polarize the change.)

Minor changes:

l. 55: clarify by writing 'Allometric slope'

l. 73: 'close relative' sounds more natural than 'closely related species'

l. 80: change 'multi-facetted' to 'multifaceted' (note: two corrections to word)

l. 94: unclear what's meant by 'fluctuation' of the growth birth. Just variation? 

l. 98-99: unclear which similarity is meant: that there's a difference similar to the diff. between male and female 3rd legs in the 1st and 2nd legs? Or just that all these legs grow faster starting in the 4th instar?

l. 104-105: comparison unclear: what's the global difference between? Male legs and female legs? Different legs within a sex?

l. 114-115: need an only: predicted to occur ONLY in individuals in good condition because.....

l. 155: change male's to males'

l. 173-175: Not clear to me on what basis one would predict that higher expression would translate into a larger RNAi effect. I could present an argument for the opposite conclusion: that with low-level expression, knockdown may reduce expression to a greater degree, leading to a stronger effect.

l. 176: check number of digits in slopes; I'm guessing this should be reported as 3.1 vs. 2.0

l. 296: change 'effect' to 'effects'

transcriptomics: this methods section omits saying what the reads were mapped against. (I assume the genome from the former companion manuscript.)

l. 339: unclear why 2nd vs. 3rd leg comparison is omitted.

l. 365: should 'wo' be 'two'?

l. 383: I was confused by 'For RNAi'; should this be 'For qPCR'?

l. 410: Please specify the version of R being run by RStudio.

l. 565: delete 'neither'

l. 575: change 'in in' to 'in'

l. 580: space needed in M. pul

l. 580: change 'states' to 'stands'

Figure 2A, Ubx female b (on graph): think value should be 0.4787

Fig. 2 and Fig. 3 graphs: When there is no significant difference between slopes, I think the results are better represented by providing a single estimate of b (or an estimate that applies to a single sex)

Supplementary materials:

Quality of some of the figures (e.g., S3 & S7); I suspect Word is to blame for this, and that either embedding some other way or using pdf may solve the problem.

---

## [Editor Report · Decision Letter 3]

25 Feb 2021

Dear Abdou,

On behalf of my colleagues and the Academic Editor, Andreas Hejnol, I'm pleased to say that we can in principle offer to publish your Research Article "The growth factor BMP11 is required for the development and evolution of a male exaggerated weapon and its associated fighting behavior in a water strider" in PLOS Biology, provided you address any remaining formatting and reporting issues. These will be detailed in an email that will follow this letter and that you will usually receive within 2-3 business days, during which time no action is required from you. Please note that we will not be able to formally accept your manuscript and schedule it for publication until you have made the required changes.

PRESS: We frequently collaborate with press offices. If your institution or institutions have a press office, please notify them about your upcoming paper at this point, to enable them to help maximise its impact. If the press office is planning to promote your findings, we would be grateful if they could coordinate with biologypress@plos.org. If you have not yet opted out of the early version process, we ask that you notify us immediately of any press plans so that we may do so on your behalf.

Thank you again for supporting Open Access publishing. We look forward to publishing your paper in PLOS Biology. 

Sincerely,

Roli

Roland G Roberts, PhD 

Senior Editor 

PLOS Biology